# Negotiable Reinforcement Learning for Pareto Optimal Sequential Decision-Making

**Nishant Desai**
Center for Human-Compatible AI
University of California, Berkeley
nishantdesai@berkeley.edu

**Andrew Critch**
Department of EECS
University of California, Berkeley
critch@berkeley.edu

**Stuart Russell**
Computer Science Division
University of California, Berkeley
russell@cs.berkeley.edu

## Abstract

It is commonly believed that an agent making decisions on behalf of two or more principals who have different utility functions should adopt a Pareto optimal policy, i.e. a policy that cannot be improved upon for one principal without making sacrifices for another. Harsanyi's theorem shows that when the principals have a common prior on the outcome distributions of all policies, a Pareto optimal policy for the agent is one that maximizes a fixed, weighted linear combination of the principals' utilities. In this paper, we derive a more precise generalization for the sequential decision setting in the case of principals with different priors on the dynamics of the environment. We refer to this generalization as the *Negotiable Reinforcement Learning* (NRL) framework. In this more general case, the relative weight given to each principal's utility should evolve over time according to how well the agent's observations conform with that principal's prior. To gain insight into the dynamics of this new framework, we implement a simple NRL agent and empirically examine its behavior in a simple environment.

## 1 Introduction

It has been argued that the first AI systems with generally super-human cognitive abilities will play a pivotal decision-making role in directing the future of civilization [Bostrom, 2014]. If that is the case, an important question will arise: *Whose values will the first super-human AI systems serve?* Since safety is a crucial consideration in developing such systems, assuming the institutions building them come to understand the risks and the time investments needed to address them [Baum, 2016], they will have a large incentive to cooperate in their design rather than racing under time-pressure to build competing systems [Armstrong et al., 2016].

Therefore, consider two nations—allies or adversaries—who must decide whether to cooperate in the deployment of an extremely powerful AI system. Implicitly or explicitly, the resulting system would have to strike compromises when conflicts arise between the wishes of those nations. How can they specify the degree to which that system would be governed by the distinctly held principles of each nation? More mundanely, suppose a couple purchases a domestic robot. How should the robot strike compromises when conflicts arise between the commands of its owners?

It is already an interesting and difficult problem to robustly align an AI system's values with those of a *single* human (or a group of humans in close agreement). Inverse reinforcement learning (IRL)

[Russell, 1998] [Ng and Russell, 2000] [Abbeel and Ng, 2004] and cooperative inverse reinforcement learning (CIRL) [Hadfield-Menell et al., 2016] represent successively realistic early approaches to this problem. But supposing some adequate solution eventually exists for aligning the values of a machine intelligence with a single human decision-making unit, how should the values of a system serving *multiple* decision-makers be "aligned"?

In the present work, we attempt to begin answering this question. We begin by observing some deficiencies of optimizing a fixed, linear weighted sum of principals' utilities, as prescribed by Harsanyi's social aggregation theorem, in the case that those principals have differing beliefs about the probability distributions dictating the agent's observations. We show that building a decision agent whose policy optimizes such an objective is not, in general, ex-ante Pareto optimal (i.e. "as evaluated before the agent has taken any actions"). Intuitively, linear weighted aggregation fails because, before the jointly owned agent has taken any actions, each principal evaluates its policy with respect to their own beliefs, meaning that a policy that selectively prefers one principal over the other conditioned on its observations can be desirable to both principals.

Section 3 addresses the shortcomings of Harsanyi-style preference aggregation by presenting the *Negotiable Reinforcement Learning* (NRL) framework. In this domain, we model each principal's prior on the environment and utility function as a Partially Observable Markov Decision Process (POMDP). We place necessary and sufficient conditions on Pareto optimality for an agent acting over these POMDPs with policy $\pi$. We then construct a third POMDP and show that the optimal policy for this single POMDP satisfies the conditions for Pareto optimality. We refer to an agent implementing a policy that solves this reduced POMDP as a NRL agent.

Following directly from this reduction is the intriguing property that a Pareto optimal policy must, over time, prefer the utility function of the principal whose beliefs are a better predictor of the agent's observations. This counter-intuitive result constitutes the main theorem of this paper. This can be seen as settling a kind of bet between the two principals: whichever principal makes the correct predictions gets to have their utility prioritized. In Section 4, we implement a simple NRL agent and make empirical observations of this bet settling behavior.

## 2 Related work

**Social choice theory.** The entirety of social choice theory and voting theory may be viewed as an attempt to specify an agreeable formal policy to enact on behalf of a group. Harsanyi's utility aggregation theorem [Harsanyi, 1980] suggests one form of solution: maximizing a linear combination of group members' utility functions. The present work shows that this solution is inappropriate when principals have different beliefs, and Theorem 4 may be viewed as an extension of Harsanyi's form that accounts simultaneously for differing priors and the prospect of future observations. Indeed, Harsanyi's form follows as a direct corollary of Theorem 4 when principals do share the same beliefs.

**Multi-agent systems.** Zhang and Shah [2014] may be considered a sequential decision-making approach to social choice: they use MDPs to represent the decisions of principals in a competitive game, and exhibit an algorithm for the principals that, if followed, arrives at a Pareto optimal Nash equilibrium satisfying a certain fairness criterion. Among the literature surveyed here, this paper is the closest to the present work in terms of its intended application: roughly speaking, achieving mutually desirable outcomes via sequential decision-making. However, the work is concerned with an ongoing interaction between the principals, rather than selecting a policy for a single agent to follow as in this paper.

**Multi-objective sequential decision-making.** There is also a good deal of work on Multi-Objective Optimization (MOO) [Tzeng and Huang, 2011], including for sequential decision-making, where solution methods have been called Multi-Objective Reinforcement Learning (MORL). For instance, Gábor et al. [1998] introduce a MORL method called Pareto Q-learning for learning a set of a Pareto optimal polices for a Multi-Objective MDP (MOMDP). Soh and Demiris [2011] define Multi-Reward Partially Observable Markov Decision Processes (MR-POMDPs), and use genetic algorithms to produce non-dominated sets of policies for them. Roijers et al. [2015] refer to the same problems as Multi-objective POMDPS (MOPOMDPs), and provide a bounded approximation method for the optimal solution set for all possible weightings of the objectives. Wang [2014] surveys MORL

methods, and contributes Multi-Objective Monte-Carlo Tree Search (MOMCTS) for discovering multiple Pareto optimal solutions to a multi-objective optimization problem.

However, none of these or related works address scenarios where the objectives are derived from principals with differing beliefs, from which the priority-shifting phenomenon of Theorem 4 arises. Differing beliefs are likely to play a key role in negotiations, so for that purpose, the formulation of multi-objective decision-making adopted here is preferable.

## 3 Negotiable Reinforcement Learning

Consider, informally, a scenario wherein two principals — perhaps individuals, companies, or states — are considering cooperating to build or otherwise obtain a machine that will then interact with an environment on their behalf.[1] In such a scenario, the principals will tend to bargain for "how much" the machine will prioritize their separate interests, so to begin, we need some way to quantify "how much" each principal is prioritized.

For instance, one might model the machine as maximizing the expected value, given its observations, of some utility function $U$ of the environment that equals a weighted sum

$$w^{(1)}U^{(1)} + w^{(2)}U^{(2)} \tag{1}$$

of the principals' individual utility functions $U^{(1)}$ and $U^{(2)}$, as Harsanyi's social aggregation theorem recommends [Harsanyi, 1980]. Then the bargaining process could focus on choosing the values of the weights $w^{(i)}$.

However, this turns out to be a bad idea. As we shall see in the following example, this solution form is not generally compatible with Pareto optimality when agents have different beliefs. Harsanyi's setting does not account for agents having different priors, nor for decisions being made sequentially, after future observations. In such a setting, we need a new form of solution, exhibited here.

**A cake-splitting scenario.** Alice (principal 1) and Bob (principal 2) are about to be presented with a cake which they can choose to split in half to share, or give entirely to one of them. They have (built or purchased) a robot that will make the cake-splitting decision on their behalf. Alice's utility function returns 0 if she gets no cake, 20 if she gets half a cake, or 30 if she gets a whole cake. Bob's utility function works similarly.

However, Alice and Bob have slightly different beliefs about how the environment works. They both agree on the state of the environment that the robot will encounter at first: a room with a cake in it ($s_1$ = "cake"). But Alice and Bob have different predictions about how the robot's sensors will perceive the cake: Alice thinks that when the robot perceives the cake, it is 90% likely to appear with a red tint ($o_1$ = "red"), and 10% likely to appear with a green tint ($o_1$ = "green"), whereas Bob believes the exact opposite. In either case, upon seeing the cake, the robot will either give Alice the entire cake ($a_1$ = (all, none)), split the cake half-and-half ($a_1$ = (half, half)), or give Bob the entire cake ($a_1$ = (none, all)). Moreover, Alice and Bob have common knowledge of all these facts.

Now, consider the following Pareto optimal policy that favors Alice (principal 1) when $o_1$ is red, and Bob (principal 2) when $o_1$ is green:

$$\hat{\pi}(- \mid \text{red}) = 100\%(\text{all, none})$$
$$\hat{\pi}(- \mid \text{green}) = 100\%(\text{none, all})$$

This policy can be viewed intuitively as a bet between Alice and Bob about the value of $o_1$, and is highly appealing to both principals:

$$\mathbb{E}^{(1)}[U^{(1)}; \hat{\pi}] = 90\%(30) + 10\%(0) = 27$$
$$\mathbb{E}^{(2)}[U^{(2)}; \hat{\pi}] = 10\%(0) + 90\%(30) = 27$$

In particular, $\hat{\pi}$ is more appealing to both Alice and Bob than an agreement to deterministically split the cake (half, half). We start to see that when principals evaluate their expected returns under a policy $\pi$ with respect to differing beliefs about action outcomes, they may mutually agree to a policy that favors one principal over the other, contingent on the action-observation history. We formalize this intuition and explore its consequences in the following sections.

### 3.1 A POMDP formulation

Let us formalize the machine's decision-making situation using the structure of a Partially Observable Markov Decision Process (POMDP) [Sondik, 1973]. It is assumed that the principals will have common knowledge of the policy $\pi = (\pi_1, \ldots, \pi_n)$ they select for the machine to implement, but that the principals may have different beliefs about how the environment works, and of course different utility functions. We refer to this as the *common knowledge assumption*.

We encode each principal $j$'s outlook as a POMDP, $D^{(j)} = (\mathcal{S}^{(j)}, \mathcal{A}, T^{(j)}, U^{(j)}, \mathcal{O}, \Omega^{(j)}, n)$, which simultaneously represents that principal's beliefs about the environment, and the principal's utility function.

$\mathcal{S}^{(j)}$ is a set of possible states $s$ of the environment.

$\mathcal{A}$ is the set of possible actions $a$ available to the NRL agent.

$T^{(j)}$ is the conditional probabilities principal $j$ believes will govern the environment state transitions, i.e., $\mathbb{P}^{(j)}(s_{i+1} \mid s_i, a_i)$.

$U^{(j)}$ is principal $j$'s utility function from sequences of environmental states $(s_1, \ldots, s_n)$ to $\mathbb{R}$. [2]

$\mathcal{O}$ is the set of possible observations $o$ of the NRL agent.

$\Omega^{(j)}$ is the conditional probabilities principal $j$ believes will govern the agent's observations, i.e., $\mathbb{P}^{(j)}(o_i \mid s_i)$.

$n$ is the time horizon.

Thus, principal $j$'s subjective probability of an outcome $(\bar{s}, \bar{o}, \bar{a})$, for any $\bar{s} \in (\mathcal{S}^{(j)})^n$, is given by a probability distribution $\mathbb{P}^{(j)}$ that takes $\pi$ as a parameter:

$$\mathbb{P}^{(j)}(\bar{s}, \bar{o}, \bar{a}; \pi) := \mathbb{P}^{(j)}(s_1) \cdot \prod_{i=1}^{n} \mathbb{P}^{(j)}(o_i \mid s_i) \, \pi(a_i \mid o_{\leq i} a_{<i}) \, \mathbb{P}^{(j)}(s_{i+1} \mid s_i, a_i) \qquad (2)$$

We say that the POMDPs $D^{(1)}$ and $D^{(2)}$ are *compatible* if any policy for one may be viewed as a policy for the other, i.e., they have the same set of actions $\mathcal{A}$ and observations $\mathcal{O}$, and the same number of time steps $n$. By the common knowledge assumption, principals' outlooks are assumed to be encoded by a set of compatible POMDPs.

### 3.2 Pareto optimal policies

In this context, where a policy $\pi$ may be evaluated relative to more than one POMDP, we use superscripts to represent which POMDP is governing the probabilities and expectations, *e.g.*,

$$\mathbb{E}_{\pi}^{(j)}[U^{(j)}] := \sum_{\bar{s} \in (\mathcal{S}^{(j)})^n} \mathbb{P}^{(j)}(\bar{s}; \pi) U^{(j)}(\bar{s})$$

represents the expectation in $D^{(j)}$ of the utility function $U^{(j)}$, assuming policy $\pi$ is followed.

**Definition 1** (Pareto optimal policies). *A policy $\pi$ is* Pareto optimal *for a compatible pair of POMDPs* $(D^{(1)}, D^{(2)})$ *if for any other policy $\pi'$, either*

$$\mathbb{E}_{\pi}^{(1)}[U^{(1)}] \geq \mathbb{E}_{\pi'}^{(1)}[U^{(1)}] \quad or \quad \mathbb{E}_{\pi}^{(2)}[U^{(2)}] \geq \mathbb{E}_{\pi'}^{(2)}[U^{(2)}].$$

It is assumed that, during negotiation, the principals will be seeking a Pareto optimal policy for the agent to follow, relative to the POMDPs $D^{(1)}$ and $D^{(2)}$ describing each principal's outlook.

If we allow the policy $\pi$ to come from the space of stochastic policies, mapping action-observation histories to distributions over actions:

$$\pi : \left( \bar{\mathcal{A}}, \bar{\mathcal{O}} \right) \mapsto \Delta\mathcal{A},$$

then we can show that the following condition is necessary and sufficient for $\pi$ to be Pareto optimal.

**Lemma 2.** *A policy $\pi^*$ is Pareto optimal to principals $1$ and $2$ if and only if there exist weights $w^1, w^2 \geq 0$ with $w_1 + w_2 = 1$ such that*

$$\pi^* \in \underset{\pi \in \Pi}{\operatorname{argmax}} \left( w^{(1)} \mathbb{E}_\pi^{(1)}[U^{(1)}] + w^{(2)} \mathbb{E}_\pi^{(2)}[U^{(2)}] \right) \tag{3}$$

*Proof.* See supplementary material. $\square$

## 3.3 Reduction to a single POMDP

We shall soon see that any Pareto optimal policy $\pi$ must favor, as time progresses, optimizing the *utility* of whichever principal's *beliefs* were a better predictor of the NRL agent's inputs. This phenomenon is most easily shown via a reduction of the POMDPs describing the outlooks of principals 1 and 2 to a third POMDP, as follows.

For any weights, $w^{(1)}, w^{(2)} \geq 0$ with $w^{(1)} + w^{(2)} = 1$, we define a new POMDP that works by flipping a $(w^{(1)}, w^{(2)})$-weighted coin, and then running $D^{(1)}$ or $D^{(2)}$ thereafter, according to the coin flip. Explicitly, we define a *POMDP mixture* as a POMDP $D = w^{(1)} D^{(1)} + w^{(2)} D^{(2)}$. We give $D$ the same action space $\mathcal{A}$ and observation space $\mathcal{O}$ as the compatible POMDPs $D^{(1)}$ and $D^{(2)}$. In the state space of $D$, we include a latent, stochastic, binary variable $B \in \{1, 2\}$ that is fixed for all time and initial belief given by $\mathbb{P}(B = 1) = w^{(1)}$ and $\mathbb{P}(B = 2) = w^{(2)}$. In the case that $B = 1$, we draw states, transition probabilities, observation probabilities, and utilities from the parameters of $D^1$. Formally,

$$\mathbb{P}_D(\bar{s}, \bar{o}, \bar{a} \mid B = 1; \pi) = \mathbb{P}^{(1)}(\bar{s}, \bar{o}, \bar{a}; \pi),$$

and

$$\mathbb{E}_\pi^{(D)}[U \mid B = 1] = \mathbb{E}_\pi^{(1)}[U^1].$$

Likewise for $B = 2$. This is a generalized version of a well-known POMDP reduction used in the Bayesian Reinforcement Learning literature [Ghavamzadeh et al., 2016].

Given any policy $\pi$, the expected payoff of $\pi$ in $D = w^1 D^1 + w^2 D^2$ is exactly

$$\mathbb{P}(B = 1) \cdot \mathbb{E}_\pi(U \mid B = 1) + \mathbb{P}(B = 2) \cdot \mathbb{E}_\pi(U \mid B = 2) \tag{4}$$

$$= w^{(1)} \mathbb{E}_\pi^{(1)}[U^{(1)}] + w^{(2)} \mathbb{E}_\pi^{(2)}[U^{(2)}] \tag{5}$$

Therefore, using the above definitions, Lemma 2 may be restated in the following equivalent form:

**Lemma 3.** *Given a pair $(D^{(1)}, D^{(2)})$ of compatible POMDPs, a policy $\pi$ is Pareto optimal for that pair if and only if there exist weights $w^{(j)}$ such that $\pi$ is an optimal policy for the single POMDP given by $w^{(1)} D^{(1)} + w^{(2)} D^{(2)}$.*

## 3.4 Structural properties of Pareto optimal POMDP reduction

Expressed in the form of Equation 3, it might not be clear how a Pareto optimal policy makes use of its observations over time, aside from storing them in memory. For example, is there any sense in which the NRL agent carries "beliefs" about the environment that it "updates" at each time step? Lemma 3 allows us to answer this and related questions by translating theorems about single POMDPs into theorems about compatible pairs of POMDPs.

We introduce the notation $h_i$ to represent the NRL agent's action-observation history at time $i$, i.e. $h_i = (o_{\leq i}, a_{<i})$. At timestep $i$, the NRL agent has a belief over the latent value $B$, conditioned on $h_i$. Note that given an action-observation history $h_i$, the NRL agent's posterior belief over the value of $B$ is proportional to the probabilities assigned by each principal's outlook to the realized observation sequence:

$$\mathbb{P}(B = j | h_i) \propto w^{(j)} \mathbb{P}^{(j)}(o_{\leq i} \mid a_{<i}).$$

This relation, combined with the expected value expression in Equation 4, reveals a pattern in how the weights on the principals' conditionally expected utilities must change over time, which is the main result of this paper:

**Theorem 4** (Pareto optimal policy recursion). *Given a pair $(D^{(1)}, D^{(2)})$ of compatible POMDPs of length $n$, a policy $\pi$ is Pareto optimal if and only if its components $\pi_i$ for $i \leq n$ satisfy the following backward recursion for some pair of weights $w^{(1)}, w^{(2)} \geq 0$ with $w^{(1)} + w^{(2)} = 1$:*

$$\pi(h_i) \in \operatorname*{argmax}_{\alpha \in \Delta A} \left( w^{(1)} \mathbb{P}^{(1)} \left( o_{\leq i} \mid a_{<i} \right) \mathbb{E}_\pi^{(1)} [U^{(1)} \mid h_i, a_i \sim \alpha] \right.$$

$$\left. + w^{(2)} \mathbb{P}^{(2)} \left( o_{\leq i} \mid a_{<i} \right) \mathbb{E}_\pi^{(2)} [U^{(2)} \mid h_i, a_i \sim \alpha] \right)$$

*In words, to achieve Pareto optimality, the machine must*

1. *use each principal's own beliefs when estimating the degree to which a decision favors that principal's utility function, and*

2. *shift the relative priorities of the principals' expected utilities in the machine's decision objective over time, by a factor proportional to how well the principals predict the machine's inputs.*

*Proof.* By Lemma 3, the Pareto-optimality of $\pi$ for $(D^{(1)}, D^{(2)})$ is equivalent to its classical optimality for $D = w^{(1)} D^{(1)} + w^{(2)} D^{(2)}$ for some $(w^{(1)}, w^{(2)})$. Writing $\mathbb{P}$ for probabilities in $D$, this is equivalent to $\alpha = \pi(h_i)$ maximizing the following expression $F(\alpha)$ for each $i \in \{0, \ldots, n\}$:

$$F(\alpha) = \mathbb{E}_\pi^{(D)} [U \mid h_i, a_i \sim \alpha]. \tag{6}$$

The above property states, in words, that the optimal policy is the one that maximizes future expected rewards given an observation history, without regard to any alternate histories and by recursively assuming the same selection process for future timesteps. This is a standard formulation for POMDPs, and is exactly Bellman's Principle of Optimality [Bellman, 1957, Chap III, 3.]

The tower property of expectation allows us to write the above expectation factor as

$$\mathbb{E}^{(D)} \pi[U \mid h_i, a_i \sim \alpha] = \mathbb{P}(B = 1 \mid h_i) \mathbb{E}^{(D)} \pi[U \mid h_i, a_i \sim \alpha, B = 1]$$
$$+ \mathbb{P}(B = 2 \mid h_i) \mathbb{E}^{(D)} \pi[U \mid h_i, a_i \sim \alpha, B = 2]. \tag{7}$$

Now, observe that, by Bayes' rule, the posterior probability

$$\mathbb{P}(B = j \mid h_i) \propto w^{(j)} \mathbb{P}^{(j)} (o_{\leq i} \mid a_{<i}). \tag{8}$$

By construction, $\mathbb{E}^{(D)} \pi[U \mid h_i, a_i \sim \alpha, B = j] = \mathbb{E}^{(j)} \pi[U^{(j)} \mid h_i, a_i \sim \alpha]$. Substituting this expression and (8) into Equation 7 gives us the result. $\square$

An intuition about this property is gained by noting that as the NRL agent takes actions in the single POMDP $w^{(1)} D^{(1)} + w^{(2)} D^{(2)}$, its posterior belief about the value of the latent variable $B$ is exactly equal to its belief about which utility function is "correct" for the POMDP it is acting in.

When the principals have the same beliefs, they always assign the same probability to the machine's inputs, so the weights on their respective expectations do not change over time. In this case, Harsanyi's utility aggregation formula is recovered as a special instance.

## 3.5 Interpretation as Bet Settling

Theorem 4 shows that a Pareto optimal policy must tend, over time, toward prioritizing the expected *utility* of whichever principal's *beliefs* best predict the machine's inputs better. From some perspectives, this is a little counterintuitive: not only must the machine gradually place more predictive weight on whichever principal's prior is a better predictor, but it must reward that principal by attending more to their utility function as well.

Thus, a machine implementing a Pareto optimal policy can be viewed as a kind of bet-settling device. If Alice is 90% sure the Four Horsemen will appear tomorrow and Bob is 80% sure they won't, it makes sense for Alice to ask—while bargaining with Bob for the machine's policy—that the machine

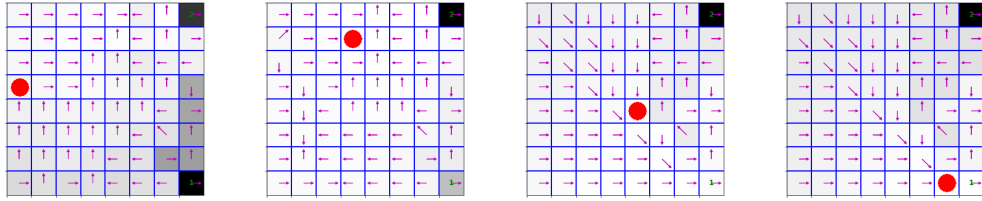

Figure 1: The agent initially heads for Goal 2 in the top-right corner. However, at Frame 9 it has observed eight deterministic transitions in a row. Its confidence in MDP 1 is high enough at that point that it veers downward and heads to Goal 1 Shown: Frames 0, 6, 12, and 15 of the trajectory.

prioritize her values more if the Four Horsemen arrive tomorrow, in exchange for prioritizing Bob's values more if they don't. Both parties will be happy with this agreement in expectation.

# 4 Evaluation

## 4.1 Experiment Environment

Our experiments are run in a modified version of the FrozenLake environment in OpenAI Gym [Brockman et al., 2016]. FrozenLake is a grid world environment that simulates a goal MDP. The agent receives a reward upon reaching the specified goal position. The agent can choose to move NORTH, SOUTH, EAST, or WEST, and the transition model can be chosen to be either stochastic or deterministic. Under a stochastic transition model, simulating the eponymous frozen lake, the agent's action fails with probability 0.2, and the agent transitions into one of the unintended neighboring positions. In the FrozenLake environment, state is fully observed by the agent. We modify the environment to support multiple possible goal states, labeled 1 and 2, corresponding to the utilities of Principal 1 and 2, respectively. For simplicity, we treat the state as fully-observable.

In these experiments Principal 1 assigns utility to the agent reaching each goal labeled 1 and has the belief that the environment has a deterministic transition model. Principal 2 assigns utility to the agent reaching each goal labeled 2 and has the belief that the environment has a stochastic transition model. The agent is initialized with initial belief state $w_1$, corresponding to a subjective belief that the agent is in Principal 1's MDP, $M_1$, with probability $w_1$ and Principal 2's MDP, $M_2$, with probability $1 - w_1 = w_2$. We use point-based value iteration to learn a the belief-space policy. The agent is then placed in either $M_1$ or $M_2$, and we observe over time as the agent acts according to its belief state.

## 4.2 Experiments

**Observed Behavior** In this set of experiments, we run the NRL agent in order to verify that its behavior resembles a type of bet-settling. The true environment is chosen to be deterministic for this experiment. After running point-based value iteration [Pineau et al., 2003] with a belief set of 331 points, we execute the resulting policy in this environment. A portion of the agent's trajectory is seen in Figure 1. The purple arrows represent the agent's choice of action at each physical position under the current belief state. The color of each square represents the agent's subjective value at each position under the current belief.

Observing the trajectory, we see that it initially moves towards goal 2. However, each time an action succeeds, the agent's belief in the stochastic environment decreases. By the ninth frame, the agent's belief in the stochastic world, and as a result its belief that goal 2 grants reward, is low enough that the policy shifts to push the agent to goal 1. This is the type of behavior we would expect of an agent that maximizes each principal's utility based on the likelihood of their beliefs.

Next, we use the same policy and place the agent in a stochastic world. The very first action results in a stochastic transition, and the likelihood of Principal 1's belief immediately falls to 0. At that point, the agent knows it is in Principal 2's MDP and believes that goal 2 will give it reward. It heads to goal 2 accordingly.

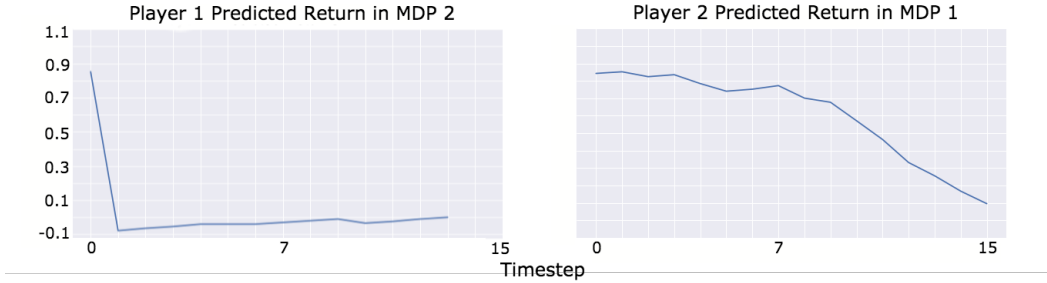

Figure 2: Left: Principal 1's predicted expected reward while in MDP 2. Right: Principal 2's predicted expected reward during trajectory depicted in MDP 1.

Full trajectories for both of these experiments are presented in the supplementary material.

**Subjective State Value During Execution**    The promise of compromise behavior may bring parties into agreement over the decision to build a NRL agent. However if, after the agent begins taking actions, parties feel that the agent will not take actions that guarantee them high future rewards, those parties may be tempted to end cooperation. The next experiment attempts to test how losing parties assess the agent's behavior during execution.

To address this concern, we turn back to the trajectories discussed in the previous section. In the first trajectory, the agent is, in fact, operating in Principal 1's MDP. At each timestep, we ask what Principal 2 believes their expected sum of future rewards to be. Recall that each principal believes the agent is acting in their own MDP. Since both principals know the agent's policy, physical state, and belief state at each timestep, each principal can estimate their future rewards by simulating the agent acting in their respective MDP with initial configuration given by the current configuration.

For the experiments here, we rollout 100 simulations from each configuration encountered along the two trajectories. For each simulation, we place the agent within the losing principal's MDP and measure the average reward attained by the agent from that starting configuration. Principal 2's assessment of their expected future rewards during execution in MDP 1 is shown in the right frame of Figure 2. Principal 1's assessment of their expected future rewards during execution in MDP 2 is shown in the left frame.

We see that until frame 9, Principal 2 believes that the agent has a high chance of reaching goal 2. It is only when the agent's behavior shifts and it veers towards goal 1 that Principal 2 begins to assess their expected future utility as decreasing sharply. In contrast, as soon as Principal 1 observes that $w_1 = 0.0$, they realize that the agent will never reach goal 1. Both of these assessments are consistent with the bet-settling behavior we expect.

## 5   Conclusion

Insofar as Theorem 4 is not particularly mathematically sophisticated—it employs only basic facts about convexity and linear algebra—this suggests there may be more low-hanging fruit to be found in the domain of "machine implementable social choice theory." To recapitulate, Theorem 4 represents two deviations from the intuition of naïve utility aggregation: to achieve Pareto optimality for principals with differing beliefs, an agent must (1) use each principal's own beliefs in evaluating how well an action will serve that principal's utility function, and (2) shift the relative priority it assigns to each principal's expected utilities over time by a factor proportional to how well that principal's beliefs predict the machine's inputs.

As a final remark, consider that social choice theory and bargaining theory were both pioneered during the Cold War, when it was particularly compelling to understand the potential for cooperation between human institutions that might behave competitively. In the coming decades, machine intelligence will likely bring many new challenges for cooperation, as well as new means to cooperate, and new reasons to do so. As such, new technical aspects of social choice and bargaining, along the lines of this paper, will likely continue to emerge.

## Footnotes

[1]The results here all generalize from two principals to $n$ principals being combined successively in any order, but for clarity of exposition, the two person case is prioritized.

[2]For the sake of generality, $U^{(j)}$ is *not assumed* to be stationary, as reward functions often are.

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
