[Supplementary Material]

# Supplementary Material for "Negotiable Reinforcement Learning for Pareto Optimal Sequential Decision-Making"

## 1 Proof of Lemma 2

**Stochastic policy assumption.** It is assumed that during the agent's first action (or before it), the agent has the ability to generate and store some random numbers in the interval $[0, 1]$, called a random seed, that will not affect the environment except through other features of its actions. Then, given any two policies $\pi$ and $\pi'$ and a scalar $p \in [0, 1]$ we may construct a third policy,

$$p\pi + (1-p)\pi',$$

that decides with probability $p$ (before receiving any inputs) to use policy $\pi$ for generating all of its future actions, and otherwise uses policy $\pi'$. (This is a "once and for all" decision; the agent does not flip-flop between $\pi$ and $\pi'$ once the decision is made.) Mixtures of more than two policies are defined similarly. With this formalism, whenever $\sum_k \alpha_k = 1$ and each $\alpha_k \geq 0$, we have

$$\mathbb{E}^{(j)}\left[U^{(j)}; \sum_k \alpha_k \pi_k\right] = \sum_k \alpha_k \mathbb{E}_{pi_k}^{(j)}[U^{(j)}]. \tag{1}$$

**Lemma 1.** *A policy $\pi^*$ is Pareto optimal to principals $1$ and $2$ if and only if there exist weights $w^{(1)}, w^{(2)} \geq 0$ with $w^{(1)} + w^{(2)} = 1$ such that*

$$\pi^* \in \underset{\pi \in \Pi}{\operatorname{argmax}} \left( w^{(1)} \mathbb{E}_{\pi}^{(1)}[U^{(1)}] + w^{(2)} \mathbb{E}_{\pi}^{(2)}[U^{(2)}] \right) \tag{2}$$

*Proof.* The stochastic policy assumption gives the space of policies $\Pi$ the structure of a convex space that the maps $\mathbb{E}_{\pi}^{(j)}[U^{(j)}]$ respect by Equation 1. This ensures that the image of the map $f : \Pi \to \mathbb{R}^2$ given by

$$f(\pi) := \left( \mathbb{E}_{\pi}^{(1)}[U^{(1)}], \ \mathbb{E}_{\pi}^{(2)}[U^{(2)}] \right)$$

is a closed, convex polytope. As such, a point $(x, y)$ lies on the Pareto boundary of image$(f)$ if and only if there exist nonnegative weights $(w^{(1)}, w^{(2)})$, not both

zero, such that

$$(x, y) \in \underset{(x^*, y^*) \in \text{image}(f)}{\text{argmax}} \left( w^{(1)} x^* + w^{(2)} y^* \right)$$

After normalizing $w^{(1)} + w^{(2)}$ to equal 1, this implies the result. $\quad\square$

## 2 NRL Agent Trajectories

In this section, we present the full trajectories taken by the agent in the deterministic and stochastic grid-world environments discussed in Section 4 of the main paper.

### 2.1 Full Trajectory in MDP 1

Depicted in Figure S1 is the agent's trajectory in MDP 1, where the environment is deterministic and Goal 1 yields utility. The agent initially heads for Goal 2 in the top-right corner. However, at Frame 9 it has observed eight deterministic transitions in a row. Its confidence in MDP 1 is high enough at that point that it veers downward and heads to Goal 1.

### 2.2 Full Trajectory in MDP 2

Depicted in Figure S2 is the agent's trajectory in MDP 2, where the environment is stochastic and Goal 2 yields utility. The agent immediately observes a stochastic transition, and its belief collapses onto MDP 2. It then moves directly to Goal 2, experiencing another stochastic transition at Frame 11.

Figure S1

# References

Figure S2