[Reviews · NeurIPS 2018]

Reviewer 1



Summary: This paper addresses the challenge of multi-objective decision making. It is an extremely relevant problem for many applications. It is often the case that a decision making agent must balance several competing objectives. They study this problem using the concept of Pareto optimality. They focus on situation where two competing principals have different beliefs on the current state. The main result is theoretical, the authors show that the agent will maximize the principal’s utility whose belief is the most accurate. Detailed comments: Overall, the paper is well-written and pleasant to read. Good high-level explanations are provided. This paper should have more experiments. The author should have chosen a POMDP environment to be more consistent with the rest of the paper. It is not really clear what is the belief of the two agents. What observation does the NRL receive? The authors provide a lot of interesting qualitative insights but the empirical verification of the results could have been stronger if carried on different POMDP environment. The NRL framework could be better motivated and applied to a more significant application than the one demonstrated here. It seems like this work could be connected to the Interactive POMDP framework. This paper and supplementary materials are missing a background section introducing the notations. Minor: - Typo in the subscripts in section 3? S1, O1, A1. (l126-127) - It would have been useful to write a formal version of Harsanyi’s theorem. - Quality of figure 2 - L251 missing figure number - Writing style in the conclusion Response to the author comments: The authors addressed my concern on the POMDP formulation of the example.

Reviewer 2



Summary: This paper reasons about a Pareto optimal social choice function in which the principles seek to agree on how to agree to use a system that acts in a sequential decision-making problem in which the principles may not share the same prior beliefs. Results suggest that to obtain such a function, the mechanism must over time make choices that favor the principle who has beliefs that appear to be more correct. Quality: The work appears to be correct as far as I have been able to discern. However, I do not like the idea of not having the proof of the main theorem (Theorem 4) in the main paper, even if for the sake of brevity. My opinion is that If the theorem is that important, its proof should be next to it. Clarity: Overall, I found the paper to be well written. However, the paper is a bit tough to follow. I did not find the math to be so difficult to follow, but the whole idea is somewhat conceptually difficult to me. I would recommend carrying examples through more thoroughly. The example given at the start of Section 3 is illuminating — I think this example could be leveraged throughout Section 3 to make the various ideas and theorems more concrete. Originality: A lot of work in multi-agent systems has dealt with computing fair equilibrium in sequential decision problems, particularly in ongoing relationships between the principles, much more than the authors acknowledge in their literature review (see for example work by Munoz and Littman, 2008 — “A polynomial-time Nash equilibrium algorithm for repeated stochastic games.” and various algorithms that cite and build on that work.) However, as the authors point out, these works deal primarily with repeated interactions between the principles, and do not consider “one-shot” sequential decision problems in which a social choice mechanism must be implemented when the players have prior beliefs. Furthermore, fair and Pareto optimal solutions in social choice are common, but I am not aware of work in this regard for sequential decision problems in which the principles have imperfect beliefs. Thus, I believe this work is novel. Significance: In some sense, I find this paper to be conceptually genius. A lot of thought went into the work, and the authors clearly see things I would not have. The ideas are quite intriguing. On the other hand, the results of the paper rely on some pretty heavy assumptions that greatly reduce the practicality of the work and hence its significance. First, the paper is based on the assumption that the principals will insist on a Pareto optimal solution. While this is an axiom sometimes used in social choice theory, I do not believe the principles would agree to this or feel good about it in this particular scenario, particularly if they are human. In practice people seem to often favor a fair a non-Pareto optimal (but more fair solution) than a Pareto optimal but unfair solution (that is unfair in the other principle’s favor). Second, the mechanism assumes knowledge of the player’s beliefs, which is impractical since the players may not want to or may not be able to communicate their beliefs in precise enough a manner to make the mechanism work as intended. Update after author response: I appreciated the author's response. My view of the strengths and weaknesses of the paper remains essentially unchanged.

Reviewer 3



** update ** I really enjoyed reading this paper, however I do not change my rating since the authors' response as well as the other reviews confirm the problems raised in mine. The first point is shared with reviewer 2: the proofs must be integrated into the paper, and the authors agree. The second point, also present in the second review, concerns how to define players' beliefs in practice: the authors propose to give adequate references dealing with this problem. Finally, I am not convinced by the authors' response to the use of the term "reinforcement learning". Their answer includes the word "Bayesian" which indeed seems to me more appropriate for the work they propose. ** summary ** In this paper the authors formalize the situation where two entities choose to cooperate sequentially in uncertainty without having the same criterion to optimize or the same model for the considered system. The mathematical models of each entity are probabilistic, Markov in time, partially observable and share the same actions and observations. After an introduction of the type of situation considered using a simple example of cake sharing, the entities' models of the system are defined as POMDPs. The definition of an Pareto-optimal policy is also given explicitly. A first result (Lemma 2.) makes it possible to identify all the optimal policies based on the expectations of gain of the entities under these policies and arbitrary weights. From this result and the definition of a POMDP having the weights mentioned above as initial belief on the true model, the main result of this paper (Theorem 4) is derived. This result makes it possible to calculate the probability distribution on the actions at each stage of the process (the Pareto-optimal strategy). The authors point out that optimization favours the entity whose model best explains the observations obtained over time, which is verified in the following experimental section. The application proposed in the experimental part sets up an agent on a grid. Each of the entities wants the agent to reach a different grid cell. One entity believes that the system is deterministic, while the other believes that agent transitions are random. Simulations, governed by each model, are performed, described and explained. ** quality ** This paper is well introduced, ideas are properly developed and interesting results are obtained. I personally find it interesting that entities have an interest in providing a model that explains reality well (observations) because it eliminates the problem of cheating (providing a model that one does not believe in order to maximize one's own utility knowing the calculations made by the jointly-owned agent). However, I find it extremely unfortunate that the proofs of the two main results do not appear in the paper but only in the supplementary materials. As noted in the conclusion, "it employs only basic facts about convexity and linear algebra" thus the authors could at least have provided a sketch of proof in order to make their work more self-contained. Moreover it seems that there is a problem of vocabulary with this paper. Since the models are known in advance and the actions are selected from calculations based on these models and not on estimates based on rewards received during tests, this work is not part of Reinforcement Learning but part of the domain of sequential decision making in uncertainty. Finally this paper exceeds the page limit (two extra lines). These three points must be changed if the paper is accepted. ** clarity ** I found this paper fairly easy to read and understand. However the following remarks should be taken into account. page 4: The formalization of the POMDP model is associated with (Smallwood "The optimal control of partially observable Markov processes over a finite horizon." Operations research 1973) rather than (Russel 2003). < P( o_i | s_i ) > P( o_{i+1} | s_{i+1} ) Otherwise, no observation for s_{n+1}. Usually, i=0:n, no observation for s_0, but an initial belief b_0) < P( s_{i+1} | s_i a_i ) > P( s_{i+1} | s_i, a_i ) < e.g. > \textit{e.g.} page 5,6: The prior distribution is often call "initial belief". page 7: < Figure > Figure 1. The design of the point-based algorithms for POMDP solving is associated with (Pineau "Point-based value iteration: An anytime algorithm for POMDPs." IJCAI 2003) rather than (Ross 2008). page 8: < is shown in the top/bottom frame > is shown in the right/left frame ** originality ** To the best of my knowledge this paper is the first to propose a solution for calculating optimal policies in the context where competing or cooperating entities have a different modeling of the system. The authors have done a good bibliographical work to position their study in the relevant literature. However, work has also been carried out with different models of uncertainty. Indeed entities may not be able to provide precise probabilities about the system under consideration and this should be mentioned. The study is limited to the case where the entities have a very precise guess about how works the system. There are other models taking into account this imprecision which is unfortunately often present in practice. For example work was produced for planning under Knightian Uncertainty, (Trevizan "Planning under Risk and Knightian Uncertainty." IJCAI 2007), with Dempster-Shafer belief Theory (Merigo "Linguistic aggregation operators for linguistic decision making based on the Dempster-Shafer theory of evidence." International Journal of Uncertainty, Fuzziness and Knowledge-Based Systems 2010), with Possibility Theory (Drougard "Structured Possibilistic Planning Using Decision Diagrams" AAAI 2014), and with Imprecise Probability Theory (Itoh "Partially observable Markov decision processes with imprecise parameters." Artificial Intelligence 2007). Some of these references should be cited since this model is intended to work in practice. ** significance ** The theoretical and application results are very interesting and make this work quite significant. It would be much more so if the proofs were present and the applications more diversified. Indeed the application is a problem with very few states and the entities' models of the system are only MDPs and not POMDPs.